# Validation of Recombinant Heparan Sulphate Reagents for CNS Repair

**DOI:** 10.3390/biology12030407

**Published:** 2023-03-04

**Authors:** Susan L. Lindsay, Rebecca Sherrard Smith, Edwin A. Yates, Colin Cartwright, Bryan E. Thacker, Jeremy E. Turnbull, Charles A. Glass, Susan C. Barnett

**Affiliations:** 1School of Infection and Immunity, Sir Graeme Davies Building, 120 University Place, University of Glasgow, Glasgow G12 8TA, UK; 2Institute of Systems, Molecules and Integrative Biology, University of Liverpool, Liverpool L69 7ZB, UK; 3TEGA Therapeutics, Inc., 3550 General Atomics Court, G02-102, San Diego, CA 92121, USA; 4Centre for Glycosciences, Keele University, Keele ST5 5BG, UK

**Keywords:** heparan sulphate, recombinant heparin mimetics, CNS repair, remyelination, neurite outgrowth

## Abstract

**Simple Summary:**

The complex tissue changes that occur after central nervous system (CNS) damage/injury require a multi-target treatment approach. Specialised sugar molecules that reside in and around our cells, known as heparan sulphate (HS), regulate numerous cellular functions and are important for tissue repair. HS contains a chemical group called sulphate which, depending on the number and position, can make cells regulate different functions. To harness the potential of HS, we can mimic its function using heparin mimetics (mHeps). mHeps are modified from the blood-thinning drug heparin whose structure resembles HS. mHeps can be chemically altered to contain less or more sulphate. We have shown that mHeps with low levels of sulphate (low-sulphated) are beneficial for CNS repair. However, mHeps are manufactured from heparin obtained from pig intestines, the supply of which can risk contamination or suffer shortages. There are also ethical implications of using animal-derived products as future therapies. Next-generation HS mimics now exist which are derived from cultured cells, termed recombinant HS mimetics (rHS). rHS may also promote repair and would translate to the clinic more readily than pig tissue-derived compounds. We have therefore validated the repair potential of these newer next-generation mimetics and compared them with our lead low-sulphated compound.

**Abstract:**

Therapies that target the multicellular pathology of central nervous system (CNS) disease/injury are urgently required. Modified non-anticoagulant heparins mimic the heparan sulphate (HS) glycan family and have been proposed as therapeutics for CNS repair since they are effective regulators of numerous cellular processes. Our *in vitro* studies have demonstrated that low-sulphated modified heparan sulphate mimetics (LS-mHeps) drive CNS repair. However, LS-mHeps are derived from pharmaceutical heparin purified from pig intestines, in a supply chain at risk of shortages and contamination. Alternatively, cellular synthesis of heparin and HS can be achieved using mammalian cell multiplex genome engineering, providing an alternative source of recombinant HS mimetics (rHS). TEGA Therapeutics (San Diego) have manufactured rHS reagents with varying degrees of sulphation and we have validated their ability to promote repair *in vitro* using models that mimic CNS injury, making comparisons to LS-mHep7, a previous lead compound. We have shown that like LS-mHep7, low-sulphated rHS compounds promote remyelination and reduce features of astrocytosis, and in contrast, highly sulphated rHS drive neurite outgrowth. Cellular production of heparin mimetics may, therefore, offer potential clinical benefits for CNS repair.

## 1. Introduction

Central nervous system (CNS) disease or injury results in complex tissue damage which leads to the loss of functional cells, and currently remains challenging to repair. There is, therefore, a need to identify new treatments that target the regulation of the multicellular processes required to repair the damaged CNS, for example, those which control remyelination, axonal outgrowth and astrocytosis. A key master regulator of multiple cellular mechanisms is the heparan sulphate (HS) glycan family. HS exists as polysaccharide chains (between 5 and 50 kDa) of repeating *N*-acetyl/*N*-sulphated D-glucosamine and a glucuronic acid (either D-glucuronic acid or its epimer, L-iduronic acid), with possible biosynthetic modifications on C2 and/or C6 by sulphation [1,2]. This range of structural diversity can influence how HS regulates a plethora of cellular signalling, physiological functions, and biological processes [2]. HS chains interact with the heparin binding domains of many proteins, including chemokines, cytokines, growth factors, proteases, adhesion molecules and lipid binding proteins and control their transport, local concentration, clearance, stability and modulate their conformation and bioactivity [1,3]. For this reason, several HS mimetics have been generated which exhibit low or no anti-coagulation activity. Examples include, Tafoxiparin for protracted labour (Dilafors, Solna, Stockholm, Sweden), CX-01 for leukemia (Cantex, Weston, FL, USA) [4] and Sevuparin for sickle cell disease (Modus, Sankt Eriksgatan, Stockholm, Sweden) [5] demonstrating the safety and tolerability of this class of drug. However, little has been published on the use of HS mimetics on the effect of neural cells in CNS injury/disease. It is known that the level of HS on oligodendrocyte precursor cells (OPCs) and the ECM can influence oligodendrocyte differentiation. For example, if HS is modified using Sulf2 an extracellular sulfatase which removes 6-*O*-sulfate from HS chains on OPCs and the ECM, then myelination is affected indirectly via WNT and BMP signalling [6]. It has also been reported that HS chains of different heparan sulphate proteoglycan (HSPG) core proteins are differently sulphated in OPCs which is thought to be functionally important as it allows the binding of growth factors such FGF-2 which interact with 2,6-*O*-sulphated residues [7]. Although our previous data have shown that treatment of purified OPCs and myelinating cultures, with LS-mHep7 does not promote OPC differentiation [8].

We have shown that a panel of non-anticoagulant HS mimetics (mHep) generated by selective desulphation of commercial heparin have beneficial effects on CNS repair mechanisms [8]. Mixed neural cell cultures generated from dissociated embryonic rat spinal cords develop robust myelinated fibers separated by nodes of Ranvier (termed myelinating cultures) [9,10]. Using these cultures, we have demonstrated that low-sulphated mHeps (LS-mHeps) promote many facets of CNS repair [11,12,13]. We have used modifications of these cultures to mimic CNS injury, which has allowed us to examine factors that regulate the process of *de novo* myelination, remyelination and neurite outgrowth after injury [8,10,14,15]. We have previously shown that as many as 108 factors are released after CNS injury that may inhibit repair. Specifically, we have shown that amyloid beta peptides (1–40/1–42) [8], CXCL10 [14] and CCL5 [16] inhibit in vitro myelination and postulate that LS-mHeps sequester or prevent these factors from binding, blunting their inhibitory effects [8]. Therefore, LS-mHeps modulate endogenous mechanisms for potential therapeutic benefit in neurodegeneration and neuroinflammation [8,11,17]. LS-mHeps are currently being tested for their efficacy to promote CNS repair in vivo with promising results [18].

Recently, we screened a range of commercially available sulphated glycomolecules (heparosans, ulvans, and fucoidans) and assessed their repair capability using CNS myelinating cultures [10]. Like heparan sulphate mimetics, these compounds mimic some facets of cellular glycosaminoglycans (GAGs). We showed that *N*-sulphated heparosans promoted myelination, whilst *O*-sulphated heparosans promoted neurite outgrowth. This work demonstrated the importance of structure to HS function. In addition, highly sulphated ulvans and fucoidans had no effect on remyelination, but CX-01, a low-sulphated porcine intestinal heparin, promoted remyelination [10]. This work highlighted the use of myelinating cultures as a screen for potential therapeutics. In this investigation, we have tested the efficacy of a new cellular source of recombinant heparan sulphates (rHS) that contain similar sulphate modifications to the well-established mHep panel. TEGA pharmaceuticals (San Diego), using mammalian cell multiplex genome engineering to direct cellular synthesis of recombinant heparin and HS, have made a library of rHS which are available commercially [19]. They have demonstrated that production of rHS using optimised bioreactors can substantially improve GAG production. rHS have high batch-to-batch consistency alongside laboratory confirmed sulphate content, chain length and purity. This has clear advantages, since currently heparin is porcine-tissue-derived and is difficult to regulate [20,21,22]. Cellular production offers an animal free source and allows the entire supply chain to be under GMP control [20]. Currently, 80% of the world’s heparin supply is sourced from China from porcine intestines [22]. Such reliance on a specific animal source of heparin comes with risk of shortage, since outbreaks of disease lead to large numbers of animals being culled [22,23]. Furthermore, animal-derived heparin products have previously been contaminated with over sulphated chondroitin sulphate, resulting in anaphylactic like reactions and deaths in the USA and other countries [23]. Presently there is serious clinical concern as to whether the global demand for heparin, which is growing inexorably, can be sustained using pigs alone [20,21,22,24]. Therefore, moving forward clinically, rHS represents a novel sustainable source for patient use.

Hence, using our established in vitro culture models of injury we have screened the repair potential of a panel of rHS with varying degrees of sulphation and compared them to our lead modified heparin compound LS-mHep7 to determine whether any show potential therapeutic benefit for the treatment of CNS disease/injury.

## 2. Materials and Methods

### 2.1. Recombinant Heparan Sulphate (rHS) Compounds

rHS was manufactured using methods published previously [19]. Briefly, CHO or MST cells were grown at 0.2 × 10^6^ cells/mL at 37 °C, 5% CO_2_ for 7 days. After centrifugation, the cell pellet was resuspended in 25 mM sodium acetate, pH 6.0, 0.25 M NaCl, 0.1% Triton X100 (wt/vol) and 0.5 mg/mL Pronase, before incubation overnight at 37 °C. Conditioned medium and protease digested cells were filtered through a 0.45 μm PES filter, and applied to 0.5 mL DEAE-Sephacel columns, equilibrated using DEAE wash buffer (25 mM sodium acetate, pH 6.0, 0.25 M NaCl). After elution using the same buffer containing 2 M NaCl, eluates were desalted on PD10 columns or by dialysis and dried by lyophilization. Dried products were reconstituted in water, digested with micrococcal nuclease overnight at 37 °C, followed by Pronase digestion for 3 h at 37 °C. Beta-elimination was performed to liberate the HS by addition of NaOH to 0.4 M with overnight incubation at 4 °C. The sample was neutralised with acetic acid, diluted with water, and reapplied to DEAE-Sephacel, before washing and eluted, as described above. Protein and DNA content were measured using BCA assay and UV absorbance. Information on rHS reagents can be found in Figure A1.

### 2.2. Low-sulphated Modified Heparin 7 (LS-mHep7)

Low-sulphated modified heparin 7 (LS-mHep7) was produced semi-synthetically by chemical selective desulphation of porcine mucosal heparin, as described (see compound **2** in [25]). The compound had the predominant disaccharide repeating structure IdoA-GlcNS and corresponding chemical desulphation at both the 2-*O* and 6-*O* positions. Information can be found in Figure 1a.

### 2.3. Astrocytes Derived from Neurospheres

Neurospheres (NS) were generated from the striata of 1-day-old SD rat pups using a method modified by [26]. They were differentiated into astrocytes as described in [9,14,27]. Briefly, neurospheres were triturated and plated onto 13 mm poly-L-lysine (PLL; 13 µg/mL, Sigma, Gillingham, UK)-coated coverslips (VWR, Leicestershire, England, UK) and incubated for 5–7 days in vitro (DIV) at 37 °C, 7% CO_2_. Neurosphere-derived astrocytes were maintained in Dulbecco’s modified Eagle’s medium containing 1 g/mL glucose (DMEM, Life Technologies, Paisley, Glasgow, UK) supplemented with 10% FBS (Sigma, Gillingham, England, UK) and 2 mM L-glutamine (Sigma, Gillingham, England, UK).

### 2.4. Astrocyte Scratch Assays

In astrocyte scratch assays, confluent monolayers of neurosphere-derived astrocytes (as described in Section 2.3) at 7 DIV were injured using a sterile P200 (200 μL) pipette tip held perpendicular to the glass, at the edge, and a scratch lesion was created spanning the full diameter of the coverslip. Post injury, all media were removed and replaced with DMEM:10% supplemented with rHS02, rHS09, rHS10 or LS-mHep7 at 100 ng/mL or DMEM:10% on its own as a control. Astrocyte monolayers were either fixed or lysed for Western blot using standard methods described below at 1 h, 2 h or 4 h post injury.

### 2.5. Myelinating Spinal Cord Cultures

Myelinating cultures form the basis of the remyelination and neurite outgrowth assays [8]. Cultures were generated from E15.5 SD rat spinal cords, which were enzymatically dissociated. Cell suspensions were plated at 150,000 cells/coverslip containing a monolayer of neurosphere-derived astrocytes (as described in Section 2.3) in the plating medium (PM) containing 50% DMEM-1 g/mL glucose, 25% horse serum (Invitrogen, Renfrew, Glasgow, UK), 25% HBSS (with Ca^2+^ and Mg^2+^, Life Technologies, Paisley, Glasgow, UK), and 2 mM L-glutamine. Cells were left to adhere for 2 h at 37 °C, and supplemented with 300 µL PM and 500 µL differentiation medium, which contained DMEM (4.5 g/mL glucose, Life Technologies, UK), 10 ng/mL biotin (Sigma, Gillingham, England, UK), 0.5% hormone mixture (1 mg/mL apotransferrin, 20 mM putrescine, 4 µM progesterone, 6 µM selenium (formulation based on N2 mix of [28] 50 nM hydrocortisone, and 0.5 mg/mL insulin known as DM+, or DM− if lacking insulin (all reagents from Sigma, Gillingham, England, UK). Each dish was fed three times a week with DM+ for 12 DIV then DM- for the following 16–20 DIV.

### 2.6. Myelinating Injured Cultures (MC-Inj)

MC-Inj cultures were generated using previously published methods [29,30]. At 24 DIV, cultures (described in Section 2.5) were cut using a 11 mm single edge razor blade (WPI, Hertfordshire, England, UK) pressed gently across the center of the coverslip. The cut created a focal cell free area (650 µm, termed a lesion) devoid of neurites and a decrease in adjacent neurite density and myelination levels. Myelination, neurite density and neurite outgrowth into the lesion could be assessed using immunocytochemistry. The cultures were treated with the panel of rHS at various concentrations at 25 DIV and allowed to recover for a further 5 DIV. Cultures were then fixed and stained as described below.

### 2.7. In Vitro Demyelinating Assays (MC-DeMy)

MC-DeMy cultures were generated based on our previously published methods [8,31]. At 24 DIV cultures (described in Section 2.5) were treated with anti-MOG (100 ng/mL, Z2 hybridoma, IgG2a [32]) and 100 μg/mL rabbit serum complement (Merck Millipore, Hertfordshire, England, UK). The following day, culture supernatants were removed treated immediately and again on day three with either DM− or DM− supplemented with rHS02, rHS09, rHS10 or LS-mHep7 at 1, 10 or 100 ng/mL or non-demyelinated cultures were left as controls. Cultures were maintained for a further 5 DIV before being fixed and stained, as described below.

### 2.8. Immunocytochemistry

Coverslips were fixed with 4% paraformaldehyde (4% PFA, Sigma, UK) for 20 min at room temperature (RT). They were then permeabilised with 0.2% Triton X-100 (Sigma, UK) at RT for 15 min, washed once then blocked with phosphate-buffered saline (PBS) with 0.2% porcine gelatin (blocking buffer, Sigma, UK) for 1 h at RT. Coverslips were stained for mature myelin using anti-MBP (rat IgG2a, 1:500, Bio-Rad, Potten End, England, UK) and axons using anti-SMI (mouse IgG1, 1:1000, Biolegend, London, England, UK) for 1 h at RT. After washing, the cultures were incubated with goat-anti mouse 568 and goat-anti rat 488 fluorophore-conjugated secondary antibodies at RT for 45 min (1:500, Life Technologies). For quantitative analysis of MC-Inj, the region adjacent to the lesion edge running 0–670 µm across the length of the coverslip was imaged at 20× magnification to allow quantification of neurite density and myelination. For quantification of neurite outgrowth into the lesion, the entire length of the injury site was imaged, except for areas that were damaged/folded. For quantification of remyelination and neurite density in MC-DeMy, images were taken at 10× magnification. For each coverslip, 10 images were taken randomly, covering the entire coverslip. In each biological repeat, coverslips were stained in triplicate; thus, 30 images were taken per condition (at least *n* = 3 biological repeats). Treated cultures were compared to non-treated injured cultures and non-treated demyelinated cultures for MC-Inj and MC-DeMy, respectively.

### 2.9. Image Analysis

Quantification of images was carried out using methods described previously [7,28]. Briefly coverslips were imaged using CellProfiler Image Analysis software (Broad Institute) [28]. For neurite density, the threshold level pixel value for SMI31 immunoreactivity (IR) was divided by the total number of pixels. The percentage of myelinated axons (PLP) was measured using CellProfiler and an in-house pipeline. All CellProfiler pipelines are available at https://github.com/muecs/cp (URL accessed on 22 September 2013). In neurite outgrowth assays (MC-Inj), the threshold level pixel value for SMI31 immunoreactivity within the lesion site was calculated as a percentage of the total pixel value of the entire lesion region/field of view.

### 2.10. Ex Vivo Spinal Cord Explant Cultures

Longitudinal ex vivo spinal cord slice cultures from P1 C57BL/6 mice were prepared as previously described [33,34,35]. In brief, spinal cords were cut longitudinally at 300 μM thickness using a McIlwain tissue chopper and transferred onto a 0.4 μm Millicell^®^-CM cell culture mesh insert in a 6-well culture plate containing 1 mL slice culture media (40% minimal essential media, 40% horse serum, 25% Earle’s balanced salt solution, 1% Pen/Strep, 1 × Glutamax and 2.6 mg/mL glucose). Cultures were fed by replacing the cell culture media every 2–3 days and maintained at 37 °C/7% CO_2_. At 14 days, demyelination was induced by 20 h incubation in 0.5 mg mL^−1^ lysolecithin, after which slices were washed twice with fresh slice culture media and treated with rHS10 or LS-mHep7 at 100 ng/mL for 14 days (the point at which spontaneous remyelination occurred). Slices were washed in PBS, and fixed with 4% PFA for 1 h at RT before being permeabilized with 100% ethanol for 20 min at −20 °C. Primary antibodies SMI-31, (1:1000, mouse IgG1, Biolegend) and MBP (1:300, rat IgG2a, BioRad) were incubated for 48 h at 4 °C in blocking buffer (1 mM HEPES, 2% heat-inactivated horse serum, 10% heat-inactivated goat serum, 1% bovine serum albumin and 0.25% TritonX in HBSS). After washing in PBS containing 0.05% TritonX, sections were incubation with goat anti mouse 568 and goat anti rat 488 fluorophore-conjugated secondary antibodies (1:500, Life Technologies) at RT for 3 h at room temperature. Images were captured at 63× magnification using a Zeiss LSM 880 confocal microscope. Myelin ensheathment (MBP), axonal density (SMI-31) was automatically quantified using CellProfiler cell image analysis software with an in-house developed pipeline slice_cultures.cpproj found at https://github.com/muecs/cp (URL accessed on 22 September 2013).

### 2.11. Western Blot Analysis

Astrocytes were lysed using CelLytic M reagent (Sigma) containing a protease inhibitor cocktail and the protein concentration determined using the NanoDrop spectrophotometer (Thermo Scientific, Leicestershire, UK). Samples were run on Tris-acetate gels (4–12%; Invitrogen, Renrew, Glasgow, UK) at 200 V for 45 min. Gels were transferred onto nitrocellulose membranes using the iBlot system (Invitrogen, Renfrew, Glasgow, UK). Membranes were incubated in blocking buffer containing 5% skimmed milk and 0.2% Triton-X100 in PBS overnight at 4 °C. GFAP (anti-rabbit, 1:10,000, Dako, Ely, England, UK) was incubated for 1 h at RT, followed by standard chemiluminescent detection. Quantification of band intensities was performed using Image J and the Gel Analysis function (Analyze > Gels > Plot Lanes), and the GFAP intensity normalized to β-actin.

### 2.12. Statistical Analysis

Parametric data are presented as means ± SEM. Differences between groups were statistically tested using the software package GraphPad Prism 6 (GraphPad Software Inc., San Diego, CA, USA). The applied statistical procedures are provided in the figure legends. The number of experimental replicates (*n*) are shown as individual data points on the graph. Each biological repeat consists of cells or spinal cords derived from an independent animal litter. Each treatment condition had 3 technical repeats and the values were averaged (10 images were collected from 3 coverslips in myelinating cultures and astrocyte scratch assays and 20 images were collected from 3 spinal cord slices). In western blot analysis, cell lysates were pooled from 3 coverslips per condition. *p* values < 0.05 were considered statistically significant. The following symbols are used to indicate the level of significance: * *p* < 0.05, ** *p* < 0.01, *** *p* < 0.001.

## 3. Results

### 3.1. Effect of rHS02, rHS09 and rHS10 on In Vitro Remyelination

Myelinating cultures were demyelinated by overnight treatment with MOG antibody and complement. The following day, cultures were treated with rHS02, rHS09 or rHS10 at 1, 10 or 100 ng/mL followed by an assessment of the level of remyelination 5 days post-treatment (Figure 1). Immediately after demyelination there was a loss of MBP wrapped axons (DeMy Dy 0), with some spontaneous remyelination over 5 days in control cultures (DeMy Dy 5) (Figure 1a). Treatment with rHS02 at 10 ng/mL and rHS10 at 1 ng/mL both significantly promoted remyelination to similar levels to that published previously for LS-mHep7 at 1 ng/mL (Figure 1b, represented by dashed line). rHS09 had no effect on remyelination with levels not different from control in all treatment concentrations tested (Figure 1a,b). There was no effect on axonal density on any of the treatments tested (Figure 1c). Comparison of the fold change (FC) increase in remyelination compared to tied day 5 control cultures, revealed that rHS10 at 1 ng/mL promoted remyelination by 2.76-fold which was comparable to the 2.68-fold increase exerted by LS-mHep7 at 1 ng/mL (Figure 1d). rHS02 promoted remyelination by 1.74-fold and rHS09 promoted remyelination by 1.61-fold suggesting that very low-sulphated compounds are more efficacious. These data suggest that low-sulphated rHS compounds have a positive effect on remyelination comparable to LS-mHep7, whereas high sulphated rHS09 had no effect.

**Figure 1 biology-12-00407-f001:**
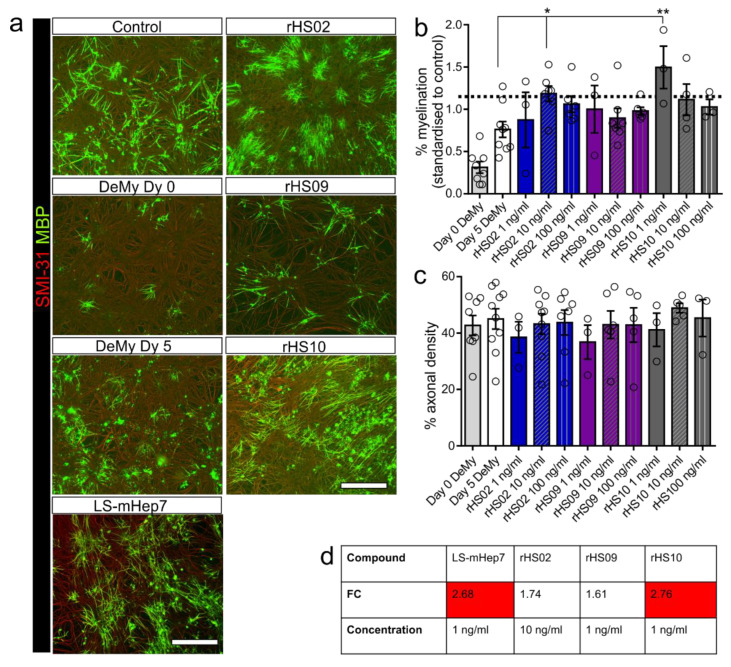
Effect of rHS02, rHS09 and rHS10 on in vitro remyelination after MOG and complement induced demyelination. Myelinating cultures at 24 DIV were demyelinated by overnight treatment with MOG (Z2) antibody and rabbit complement (termed MC-DeMy after demyelination). The following day all media was removed, and cultures treated with either, rHS02, rHS09 or rHS10 at 1, 10 or 100 ng/mL or left untreated as control. 5 DIV post treatment, cultures were fixed and stained to assess the level of remyelination. (**a**) Representative images of control untreated cultures, those after overnight demyelination (DeMy Dy 0), or 5 DIV post demyelination (DeMy Dy 5), or after 5 DIV post treated with LS-mHep7, rHS02, rHS09 and rHS10 at 10 ng/mL. SMI-31 stains axons in red, MBP stains myelin sheath in green. Scale bar represents 50 μm. (**b**) Graph shows quantification of % myelination. rHS02 at 10 ng/mL and rHS10 at 1 ng/mL significantly promoted remyelination. The average level of remyelination produced by LS-mHep7 at 1 ng/mL is illustrated by the demarcated dashed line. (**c**) Graph shows quantification of % axonal density showed no significant effects of any of the compounds tested. (**d**) Table details the average FC in remyelination compared to Day 5 DeMy cultures. LS-mHep7 and rHS10 both increase remyelination to a similar level. (*n* number represented by individual circle data points on the graph, ANOVA with Dunnett’s post-test, * *p* < 0.05, ** *p* < 0.01).

### 3.2. Effect of rHS10 on Ex Vivo Slice Culture Remyelination

To further examine the ability of rHS10 to promote remyelination, it was tested in more complex ex vivo spinal cord slice cultures. At 12 DIV, slice cultures were demyelinated overnight using lysolecithin. The following day, slices were treated with LS-mHep7 or rHS10 at 100 ng/mL or fed in standard culture media as a control. At 14 DIV the level of remyelination was quantified. After demyelination there was a loss of MBP wrapped axons (Dy 0 DeMy), with some spontaneous remyelination over 14 days in control cultures (Dy 14 Demy) (Figure 2a,b). Treatment with rHS10 significantly promoted remyelination to a similar extent as LS-mHep7 (Figure 2a,b). There were no significant differences in the axonal density between cultures (Figure 2c). These data confirm that rHS10 promotes remyelination like LS-mHep7 and suggests that *N*-sulphation is an important facet for promoting CNS myelination following injury.

### 3.3. Effect of rHS02, rHS09 and rHS10 and LS-mHep7 on Neurite Outgrowth and Lesion Width after Injury

Myelinating cultures were mechanically injured using a scalpel blade at 24 DIV (MC-Inj) to create a cell free area and immediately treated with rHS02, rHS09, rHS10 or LS-mHep7 at either 1 or 10 ng/mL. After 5 DIV, assessments of neurite outgrowth or lesion gap width were made (Figure 3). LS-mHep7 at 10 ng/mL was found to significantly promote neurite extension into the injury lesion gap compared to the non-treated control injured cultures (Figure 3a,b). In addition, rHS09 at 10 ng/mL, similarly to LS-mHep7, significantly promoted neurite outgrowth compared to control injured cultures (Figure 3a,b). Measurements of injury gap width revealed that LS-mHep7 and rHS09 at 10 ng/mL had correspondingly reduced gap width, suggesting that both these compounds had beneficial effects on neurite extension and prevention of injury site expansion (Figure 3a,c). rHS10 at 1 ng/mL also significantly reduced the injury area width compared to control injured cultures (Figure 3c). Comparisons of FC in neurite outgrowth compared to tied cut control cultures revealed that rHS09 promoted neurite outgrowth 14.56-fold, which was comparable to LS-mHep7 at 12.40-fold. Both rHS09 and LS-mHep7 also had similar reductions in average injury gap width reductions compared with controls of 60.29% and 61.47%, respectively.

After injury, myelination adjacent to the lesion edge is reduced; therefore, the amount of remyelination after treatment with rHS02, rHS09, rHS10 or LS-mHep7 was quantified (Figure 3d). There were no differences in the level of remyelination in any of the treatments tested, although LS-mHep7 had the highest levels in general. These data suggest that highly sulphated rHS09 is the most efficacious in promoting neurite outgrowth, implying that *O*–sulphation is essential for neurite outgrowth effects.

### 3.4. Effect of rHS02, rHS09 and rHS10 and LS-mHep7 on Astrocyte Reactivity

Monolayers of astrocytes were injured by scratching using a pipette tip. Immediately after injury, cells were treated with rHS02, rHS09, rHS10 or LS-mHep7 at 100 ng/mL (Figure 4a). Measurements of scratch width were made at 1, 2, 4 and 24 h. It was found that like LS-mHep7, rHS10 promoted a faster recovery of astrocyte gap closure at 4 h compared to untreated astrocytes (Figure 4b,c). Although only LS-mHep7 promoted closure compared with control untreated cultures at 24 h (Figure 4d), there were no significant differences between the % gap width of LS-mHep7, rHS02 and rHS10, suggesting a similar rate of closure (Figure 4d). Conversely, treatment with rHS09 prevented astrocyte scratch closure with a significantly wider gap width compared to control at both 4 and 24 h (Figure 4c,d). Quantification of GFAP immunoreactivity immediately adjacent to the lesion edge using thresholding revealed an increase in GFAP expression 1 and 2 h after injury in control injured cultures (Figure 4f). This increased expression post scratch was not found in LS-mHep7, rHS02 or rHS10 treated cultures with a significant reduction in GFAP staining compared to control 1 h post-injury in LS-mHep7 and rHS10 treated cultures. This suggests a dampening of GFAP reactivity after injury at the early time point with low-sulphated compounds (Figure 4g,h). Interestingly, rHS09 was found to increase GFAP immunoreactivity to similar levels as control (Figure 4g,h). Quantification of global GFAP expression by Western blot showed that after injury, there was a significant increase in GFAP expression at 1 and 2 h after injury compared with control uninjured cultures (Figure 4e,f). However, there were no significant changes in any of the treatments compared to either uninjured or cut astrocytes, suggesting the changes in GFAP expression are more localised to the lesion edge rather than the entire astrocyte population.

## 4. Discussion

HS mimetics are attractive biochemical tools and potential therapeutics due to their ability to interact with and modulate a large range of cellular processes [36]. We have previously shown that LS-mHep7 can exert beneficial effects on mechanisms which promote repair after injury including astrocytosis, myelination and neurite outgrowth and represent a novel candidate therapeutic [8,17]. However, LS-mHep7 is derived from pharmaceutical heparin purified from pig intestines, in a supply chain that is at risk of shortages and contamination [20]. There are also the clear ethical implications of using animal-derived medications for certain religious groups or fractions of society. In this investigation, we have tested a panel of next-generation which have been produced by cells with modified biosynthetic pathways as an alternative to animal-derived heparin sources [19,20]. This, therefore, has the potential to be a clinically relevant source of HS mimetic, which would allow an easier transition to the clinic. Here, we used several modifications of myelinating cultures that mimic CNS injury, to screen a panel of rHS to determine whether the level of sulphation effected their potential to promote re/myelination, neurite outgrowth or characteristics of astrogliosis making comparison to LS-mHep7.

### 4.1. Low-Sulphated rHS Promote Remyelination

We have previously shown that the sulphation level or its position on the HS disaccharide is crucial in regulating cellular function [8,11]. LS-mHeps promote neurite outgrowth and remyelination after injury, although they do not promote *de novo* myelination [8]. This is likely to be because endogenous HS are sufficient for developmental myelination, but during injury there are secreted factors present in the extracellular injury environment that LS-mHeps sequester or interact with, aiding repair [8]. It was found that rHS10, which has similar disaccharide composition and sulphate composition to the well-established LS-mHep7, efficiently promoted remyelination to comparable levels in vitro. Further investigation using more complex ex vivo slice explant cultures verified the abilities of both rHS10 and LS-mHep7 to promote remyelination after lysolecithin-induced demyelination. This is the first demonstration of the application of low-sulphated rHS for CNS repair and suggests that *N*-sulphation is important for promoting remyelination after injury since both these reagents are predominantly *N*-sulphated. *N*-sulphation is already known to be essential in mediating specific protein interactions [37]. In addition, rHS02 was found to significantly promote in vitro remyelination, but was less efficacious than rHS10 as shown by FC analysis. rHS02 is also *N*-sulphated but to a lesser degree and contains more 6-*O-*sulphate groups than either rHS10 or LS-mHep7. The lesser effect of rHS02 could again highlight the importance of *N*-sulphation in terms of remyelination potential. rHS09 which is highly sulphated, containing *N*-, 2-*O* and 6-*O* sulphate groups, had no effect on remyelination. We have shown previously that the marine glycomolecules, fucoidans and ulvans, with underlaying structures distinct from HS, also do not promote in vitro remyelination [10]. This highlights the importance of the HS backbone in LS-mHep7 and rHS10 to their activities during remyelination. Indeed, HS backbone modifications are pivotal to HS-protein interactions, with select sulphation positions and uronic acid epimerisation forming potential protein binding sites. For example, 6-*O* sulphation has been shown to be fundamental in Wnt signalling [38,39], which regulates numerous aspects of neural precursor development, as demonstrated by knockdown of the heparan sulphate 6-*O*-sulphotransferase gene, resulting in misregulated Wnt signalling [40,41]. Furthermore, structure-activity relationship studies using a combination of HS oligosaccharide libraries and FGF2 affinity chromatography, identified the role of 2-*O*-sulphated uronic acid and *N*-sulphated glucosamine in ligand binding [42]. Conversely, 6-*O*-sulphate moieties are non-essential for HS-FGF2 binding, but are required for the accompanying receptor interaction [43].

### 4.2. High-Sulphated rHS Promote Neurite Outgrowth

Previously we have shown that heparins bearing one sulphate substitution per repeating disaccharide unit at either the 2-*O*- or *N*-sulphated positions promote neurite outgrowth [8]. Caenorhabditis elegans mutants which lack the enzymatic activity of 2-*O*-sulphation, display axonal patterning defects, confirming the importance of 2-*O* sulphation in neurite outgrowth and pathfinding [44]. However, other mHep mimetics which lack any sulphated moiety in the three modification positions also promote neurite outgrowth [8], although it cannot be discounted that the negative charge provided by the uronic acid may provide the necessary binding to target proteins. Others have shown that HS with different sulphation modifications disrupts axons guidance in the Xenopus visual system, with 2-*O*- and 6-*O*-sulphated HS showing marked effects [45]. Therefore, it has been postulated that there is a sulphation code that regulates axon guidance [46]. Here, we found that only the highly sulphated rHS09 promoted neurite outgrowth. Similar to a previous investigation, we showed that the highly sulphated epimerized heparosan, Epi K5 OS (H), promoted neurite outgrowth [10]. This implies that *O*-sulphation and the epimerisation state of the uronic acid may be key for effecting neurite outgrowth. Indeed, the presence of *O*-sulphate moieties has been deemed essential to the activity of K5 derivatives as FGF signalling inhibitors [47]. Furthermore, there exists an important interplay between HSPGs and chondroitin sulphate proteoglycans (CSPGs); the balance of HS-GAG and CS-GAG binding to the PTPsigma receptor present on both CSPGs and HSPGs influences axon regeneration [48]. Using a series of HS octasaccharides it was found that HS containing at least one sulphate would bind and cancel the inhibitory effect of CSPGs and permit neurite outgrowth [49]. Thus, the contribution of CSPGs present which inhibit neurite extension through the ECM are an important consideration for HS mimetic function. Any HS mimetic that modulates the effects or amounts of chondroitin 6-sulphate and/or chondroitin-4-sulphate will enhance axon regeneration [48].

### 4.3. Astrocyte Reactivity

Astrocytes become reactive and secrete inflammatory molecules that modify the environment after CNS injury or disease [47,50,51,52]. HSPGs are found in the brain and are expressed by both neurons and astrocytes [53], although the role they play may be different [54]. Following rat brain injury there is an increase in HSPGs around the injury site and increased levels of 2-*O*-sulfated HS expressed by astrocytes [7]. We have previoulsy found little evidence of astrocyte reactivity modulation following LS-mHep7 treatment in astrocyte scratch assays assessed by Western blots [8], also true for Western blot analysis carried out in this investigation. However, thresholding analysis of GFAP staining intensity immediately at the lesion edge showed that the low-sulphated compounds LS-mHep7 and rHS10 reduced GFAP immunoreactivity at early time points. In addition, measurement of scratch width revealed that both compounds promoted astrocyte scratch closure. A proposed mechanism could be via their interaction with FGFs, which are known to modulate astrocytic outgrowth and scar formation [13]. Conversely, the highly sulphated rHS09 prevented astrocyte closure and increased GFAP reactivity at the immediate lesion edge. We have demonstrated that highly sulphated heparins or oversulphated heparin mimetics induced strong boundaries between olfactory ensheathing cells (OECs) and astrocytes, while lower sulphated heparin variants allowed cells to mingle [11,13]. Interestingly, boundary formation was dependent on *O*-sulphation, since *N*-acetylated heparin (in which *N*-sulphates are replaced with *N*-acetyl groups) also induced boundary formation [11]. Therefore, astrocyte reactivity appears to be modulated by sulphation level since low-sulphated rHS10 have similar beneficial effects as LS-mHep7.

Since MC-DeMy and MC-Inj contain neurosphere-derived astrocytes as a supportive monolayer, rHS compounds could modulate the astrocytic response post injury in these models. We have previously shown that astrocytes upregulate GFAP and CSPG expression in the injury site of MC-Inj after cutting [30]. However, none of the low-sulphated rHS compounds which reduced astrocyte reactivity also promoted neurite outgrowth, which contrasts with LS-mHep7 which does promote neurite outgrowth and reduce astrocyte reactivity. This may suggest that the repair mechanisms of rHS compounds in more complex cultures target different cell types. Myelinating cultures contain numerous CNS cells, including microglia, radial-glia, neurites, OPCs meaning the injury environment is of greater complexity than astrocytes alone. The potential effect of rHS compounds on the CNS cell types present within MC-DeMy and MC-Inj is yet to be explored and requires further investigation.

### 4.4. Molecular Weight

An important consideration is whether the molecular weight (MW) of rHS could be important for their therapeutic action. The degree of oligosaccharide polymerisation (dp) is directly related to its MW and thought to be a key feature affecting activity, certainly in relation to chemokine activity [55]. A minimum dp is required for specific protein binding, for example a tetrasaccharide (dp4) is required for FGF2 binding and the formation of its ternary complex [56,57]. However, it has been demonstrated that low MW heparin is more effective at reducing tumour growth, by inhibiting endothelial cell proliferation, than unfractionated heparin [58]. Thus, the approximate MW of rHS02 and rHS10 being almost four times, and rHS09 being twice as large as LS-mHep7 could potentially explain why LS-mHep7 appears to be the most efficacious for repair across the myelinating culture screens. Furthermore, we found that rHS10 was most efficacious at promtoing remyelination at 1 ng/mL but had less ability at higher doses. This could also be related to its MW and greater negative charge, making it less capable of interacting with secreted factors or interacting with the lipid bilayer of cells, such as OPCs at higher concentrations. Shorter chain 6-*O-*phosphorylated HS oligosaccharides have recently been shown to inhibit amyloid β aggregation in vitro [59]. The findings suggest that small sulphated glycans could competitively interfere with interactions between HS/heparin and Aβ, preventing amyloid plaque formation. Their short saccharide chain length prevents them from acting as scaffolds. We, too, have shown that LS-mHep7 modulates the degradation of Aβ peptides (1–40/1–42) by sequestering or inhibiting their aggregation following demyelination [8]. Therefore, the importance of MW in terms of the repair potential of rHS compounds requires further investigation.

## 5. Conclusions

In conclusion, we have shown that rHS-derived mimetics are beneficial for CNS repair. In particular, the low-sulphated compound, rHS10, promoted remyelination and modulated astrocyte reactivity similar to LS-mHep7, whereas the highly sulphated compound, rHS09, promoted neurite outgrowth. This confirms the importance of HS sulphation for CNS repair and provides excellent preliminary evidence of the in vitro repair benefits of rHS compounds. Future translation to the clinic will require evidence that rHS can affect repair in vivo using models of CNS injury and/or disease. However, since rHS are larger in MW and more negatively charged than LS-mHep7, alterations in their delivery such as intracerebroventricular injection (ICV) may be required, or, alternatively, their manufacture optimised to allow them to cross the blood–brain barrier more easily.

## 6. Patents

The following patents are owned by Jerry E Turnbull, Ed A Yates, and Susan C Barnett: Agents for the prevention and/or treatment of central nervous system damage (UK 1,219,696.0 and US 14/440,005). The following patents are owned by TEGA Therapeutics: in vitro heparin and heparan sulfate compositions and methods of making and using (PCT/US2017/066860). Heparin and heparan sulfate from modified MST cells and methods of making and using (PCT/US2020/048243). Cellular glycosaminoglycan compositions and methods of making and using (PCT/US2016/067373).

## Figures and Tables

**Figure 2 biology-12-00407-f002:**
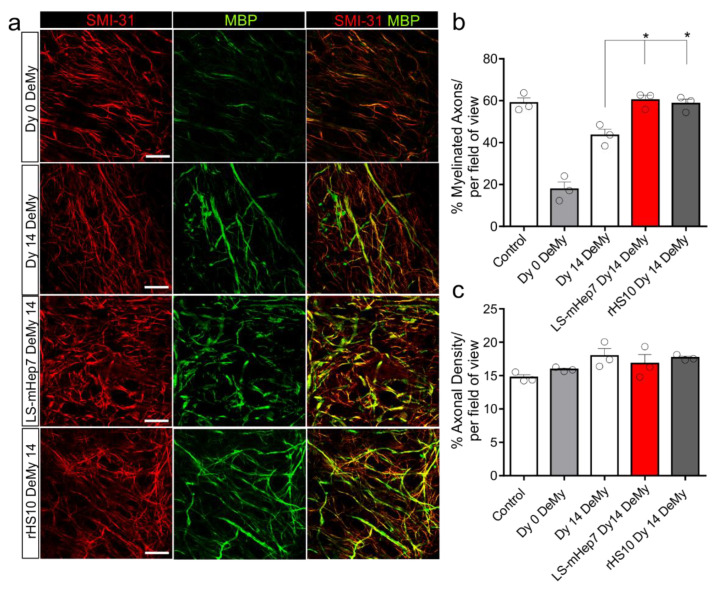
rHS10 and LS-mHep7 enhances remyelination after lysolecithin induced demyelination of ex vivo spinal cord slice cultures. Spinal cord slice cultures after 14 DIV were demyelinated using lysolecithin for 20 h (Dy 0 DeMy). Slices were then treated with either LS-mHep7 or rHS10 at 100 ng/mL or in standard culture media as a control every other day for further 14 DIV. (**a**) Representative images of slice cultures. MBP stains myelin in green and SMI-31 stains axons in red. Scale bar represents 100 μm. (**b**) The percentage of myelinated axons per field of view was enhanced significantly in both LS-mHep7 (LS-mHep7 DeMy 14) and rHS10 (rHS10 Dy 14 DeMy) treated slices compared to untreated controls (Dy 14 DeMy) (**c**). Axonal density per field of view remained unchanged across the treatments. (n number represented by individual circle data points. ANOVA with Dunnett multiple comparison * *p* < 0.05).

**Figure 3 biology-12-00407-f003:**
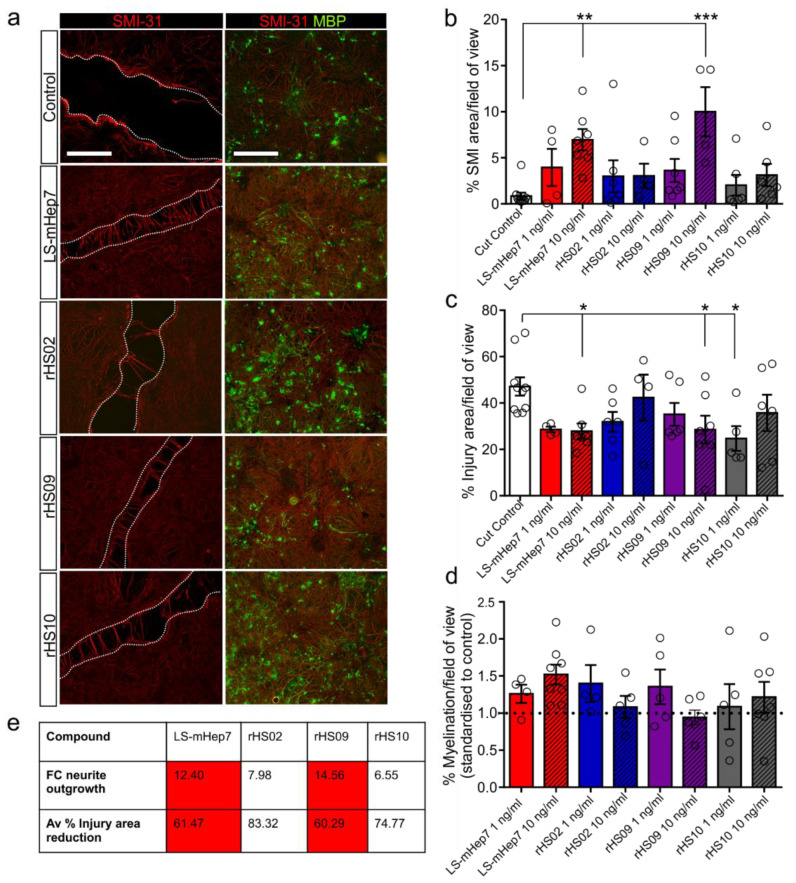
Effect of LS-mHep7, rHS02, rHS09 and rHS10 on neurite outgrowth and myelination after injury. Myelinating cultures at 24 DIV were injured by cutting with a scalpel blade (termed MC-Inj after injury). Cultures were immediately treated with LS-mHep7, rHS02, rHS09 or rHS10 at 1 or 10 ng/mL or left untreated as control. 5 DIV post treatment, cultures were fixed and stained to assess both neurite outgrowth across the lesion, gap width or myelination adjacent to the lesion. (**a**) Representative images of control cut, or myelination adjacent to the lesion. LS-mHep7, rHS02, rHS09 and rHS10 at 10 ng/mL. SMI-31 stains axons in red and MBP stains myelin in green. Dotted line demarcates injury site. Scale bar represents 50 μm. (**b**) Graph shows quantification of % axonal outgrowth in the injury region of interest (ROI). Both LS-mHep7 and rHS09 at 10 ng/mL significantly promoted neurite outgrowth across the lesion compared to control cultures. (**c**) Quantification of injury gap width. (**d**) Quantification of % myelination adjacent to the lesion. (**e**) Table details the average FC in neurite outgrowth and the average % reduction in injury gap width compared to tied cut control cultures. Red highlights similarity in FC differences between LS-mHep7 and rHS09 (*n* number represented by individual circle data points on graph, ANOVA with Dunnett’s post-test, * *p* < 0.05, ** *p* < 0.01, *** *p* < 0.001).

**Figure 4 biology-12-00407-f004:**
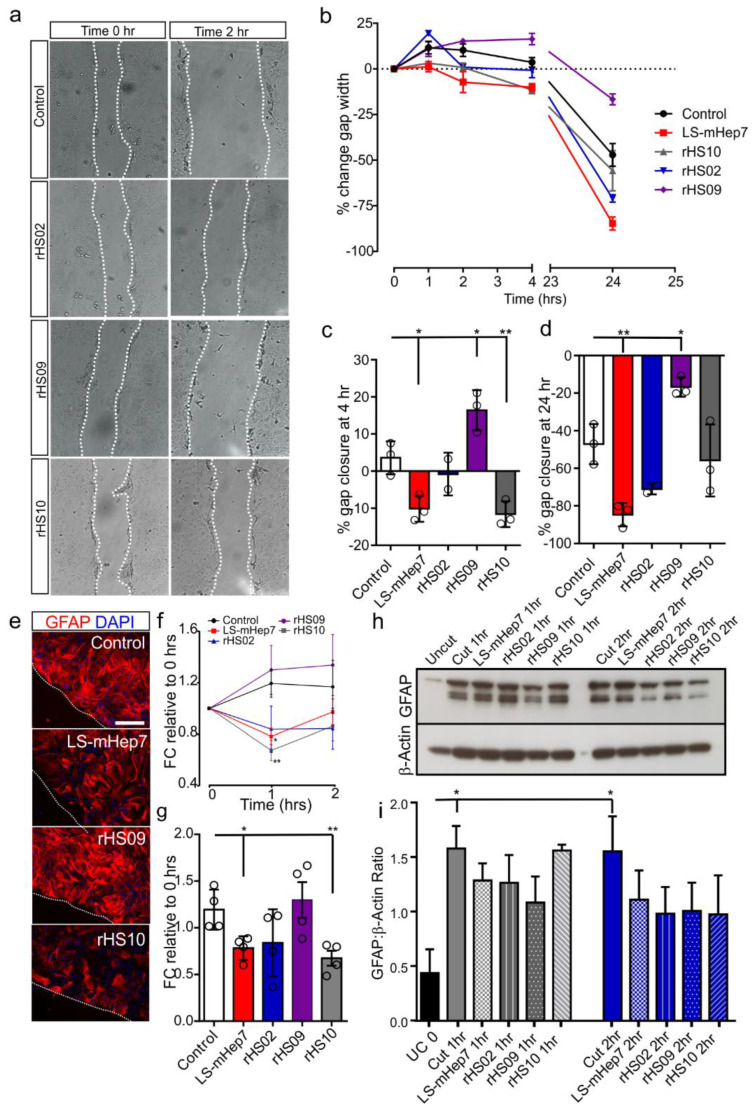
Effect of LS-mHep7, rHS02, rHS09 and rHS10 on astrocyte reactivity after injury. Monolayers of astrocytes were injured by scratching using a pipette tip down the center. Immediately after injury, cells were treated with rHS02, rHS09, rHS10 or LS-mHep7 at 100 ng/mL) or media as a control. (**a**) Representative phase images of scratch astrocytes immediately after injury (Time 0 h) or 2 h after injury (Time 2 h) in control, rHS02, rHS09 or rHS10 treatment (**b**) Measurements of scratch width were made on phase images at 1, 2, 4 and 24 h following the same injury site over time and gap width is presented as a % of the original width at time 0. (**c**) LS-mHep7 and rHS10 promoted a faster closure of gap width at the 4 h timepoint compared to control astrocytes, whereas rHS09 caused the gap to widen (**d**) LS-mHep7 promoted a faster recovery of astrocyte gap closure at 24 h compared to untreated astrocytes, whereas rHS09 significantly prevented closure. (**e**) Representative immunohistochemical images of GFAP immunoreactivity immediately adjacent to the lesion in control cut cultures and those treated with LS-mHep7, rHS09 or rHS10, 1 h post-injury. Dashed line indicates injury edge. GFAP stains astrocytes red, DAPI stain nuclei in blue. Scale bar represents 50 μm. (**f**) Quantification of GFAP immunohistochemistry using Image J thresholding of images captured at the lesion edge at 1 and 2 h post injury, showed an increased in GFAP immunoreactivity in control astrocytes at the lesion edge which was not found in LS-mHep7, rHS02 and rHS10 treated astrocytes. (**g**) There was a significant reduction in GFAP staining in LS-mHep7 and rHS10 treated astrocytes at 1 h post injury. rHS09 caused a similar increase in GFAP at 1 and 2 h to that found in control injured astrocytes. (*n* number represented by individual data points on the graph, ANOVA with Dunnett’s post-test, * *p* < 0.05, ** *p* < 0.01). (**h**) Western blot analysis of global GFAP levels in cut astrocytes treated with rHS02, rHS09, rHS10 or LS-mHep7 lysed at 1 and 2 h post injury. β-actin was used as a loading control. (**i**) Quantification showed there was a significant upregulation of GFAP 1 h and 2 h post injury in cut cultures compared to control. There was no difference in the level of GFAP expression in any treatment at any of the time points analysed when compared to untreated cut cultures. (*n* = 4, Two-way ANOVA, Dunnett’s post-test, * *p* < 0.05).

## Data Availability

Raw data were generated at the University of Glasgow. Derived data supporting the findings of this study are available from the corresponding author [SLL] upon reasonable request.

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
