# Peer review of "Validation of Recombinant Heparan Sulphate Reagents for CNS Repair"

_biology, 2023, doi:10.3390/biology12030407_

Round 1
Reviewer 1 Report
Authors validated the CNS repair potential of recombinant HS mimetics (rHS) derived from cultured cells compared to their lead low sulphated heparin mimetics (LS-mHeps) using in vitro culture models.
These data need to be confirmed in vivo in an animal model such as EAE or demyelination animal model in subsequent studies.
Introduction
Line 88: Please mention the factors and write few sentences indicating their CNS repair mechanisms.
Materials and Methods
Line 172: Please use the abbreviation only "DS" of Sprague Dawley.
Results
Table 1d. If the abbreviation of LS-mHep7 is H7, please add that in the figure legend as an abbreviation.
Line 315, 362 & 459: Please correct the scale bar.
Line 319 & 368: Please replace fold change (FC) with FC.
Figures and tables: Please unify the symbol of LS-mHep7. If it is H7, please unify it on the images and in the tables
Figure legends: Please remove sentences explaining the data, they are already mentioned in the text.
Line 369: please remove “compared to”.
Discussion
I suggest summarizing the first paragraph, many of details were mentioned in the introduction section.
Expressions abbreviated in the introduction section should be used as abbreviations in the following sections of manuscript. Some expressions and their abbreviations are repeated in the discussion section.
Conclusions
Authors need to add in the conclusion section few sentences about the future studies needed to confirm the biological activity of these mimetics for CNS repair such as in vivo studies in EAE and/or demyelination animal models and how this will be a translational application of these mimetics in treating the demyelinating diseases such as multiple sclerosis.
Reviewer 2 Report
This study addresses the use of recombinant heparan sulphate mimetics with varying degrees of sulphation in tissue repair. Thereby, low-sulphated rHS promote remyelination and reduce astrocytosis, while highly sulphated rHS drive neurite outgrowth. Facing a possible shortage a of porcine derived HS supply (as indicated by the authors) for medical treatments, it is necessary to test other sources, such as recombinant HS, rendering the study very relevant. It is also interesting to study the effect on cells of the CNS, as this may open up new options for therapies. The study design is straight forward.
I have the following comments/questions:
Introduction:
General: The authors clearly already did a lot of work in the field of heparan sulphate. The introduction would benefit from inserting additional work of other groups about the use of heparan sulfates and their effect on different cell types within and outside the CNS, if possible.
The last sentence is a bit vague. Please rephrase and be more specific.
Methods:
What do you mean by astrocyte differentiation? Can you please specify the astrocytes that are used for the experiments?
Is there a specific reason to go via neurospheres as compared to just seed homogenized cortices in order to get astrocytes in culture?
Why did you use so many different sera for the spinal cord explant cultures?
Results:
3.1
Line 289: promoted what? Is this all 1ng? Otherwise the conclusion about rHS02 would not be consistent (the authors write in line 282/282 that rHS02 is similar to HS 10 and 7, but it is clearly lower for 1 ng).
Figure 1: it is written in the figure legend that LS-mHep7 was used as a control, but not shown in the images. The images showing the effect should be added to obtain a side by side comparison.
Do you have an explanation why rHS 10 10ng less efficient as 1 ng?
Line 315: scale bar is not indicated correctly
N number: biological or technical replicates?
3.2
Is there a reason why different demyelination protocols were used? Is the antibody-induced demyelination not efficient in spinal cord explants?
Fig 2: Why is the concentration of 100ng/ml used for the compounds in this approach?
I am puzzled by the various concentrations used for the different experiments. Could you please explain why you used different concentrations for the different experiments?
3.3
Fig 3: How do you explain the difference in the effect on myelination the different compounds exert? While rHS02 1ng is more efficient in Fig 3 as compared to the 10 ng, it is the other way round in Fig 1. While the rHS10 1 ng is most efficient in Fig 1, it is not in Fig 3, and so on, apart from the fact that there are no significant differences in Fig 3 concerning myelination at all. Are there not enough axons to be myelinated? Is myelin degenerated after the scratch injury?
Line 370: what shows? Is there a word missing?
3.4
Considering the contribution of astrocytes: how can you be sure that the effects of rHS that you describe is not mediated through astrocytes, since your cultured cells were seeded on astrocytic layers?
This could be clarified by seeding cultures of purified neurons and/or oligodendroglial cells onto substrates without astrocytes, to figure out whether astrocytes really do not directly influence the observed cellular effects and what effect the rHS has on each cell type. Apparently, the HS compounds block reactive astrocytes (as described in other papers of the group) and have a direct effect on astrocytes, as shown in this study.
Discussion:
Line 494/495: this sentence is weird, is there a word missing?
Line 498: typo in the word ‘completely’
Line 506: in which paper has it been shown that LS-mHeps does not promote de novo myelination?
How do you differentiate between developmental myelination and remyelination after injury in your models using cells from very young animals (around birth)? Which cell type do you think secrets factors (did you test which factors are secreted?) contributing to the ‘injury environment’ (Line 508)?
Regarding the effects of Wnt (Line 528), does Wnt itself influence astrocyte behavior, remyelination or neurite outgrowth?
Please comment on the effect of HS on the oligodendroglial lineage?
Line 557: what is C4 and C6S?
Line 569: true of or true for?
In your section about Astrocyte reactivity (4.3): This will also impact the other cell types involved in regeneration. Could you comment on the effects altered astrocyte reactivity has in your models?
When indicating the use of rHS in CNS repair, will these compounds be able to cross the BBB? Are there any studies or did you test it yourself?
Are there already any in vivo studies using rHS or modified HS targeting CNS injury?
Reviewer 3 Report
This is a well written manuscript describing the effects of HS mimetics on neurite outgrowth and re-mylination of developing axons after injury using in vitro models. It builds upon previous work nicely. There are no major issues with this work and only minor grammatical errors, punctuation and spelling errors were detected (See attached pdf). Also some parts of the methods need a little more detail.
One area that could be considered for improvements is the repetition of detail in the results text vs the figure legends. In some cases they are almost identical.
It will be interesting to see how this work develops into mature models of CNS injury and whether these compounds will work in vivo.

Round 2
Reviewer 2 Report
The authors have addressed all questions raised during report 1. I think it is a well written manuscript with interesting findings concerning the use of HS as therapeutic agent.